# Epidemiological Overview of Urogenital Gonorrhea in Mexico (2003–2020)

**DOI:** 10.3390/healthcare11152118

**Published:** 2023-07-25

**Authors:** Miguel Ángel Loyola-Cruz, Verónica Fernández-Sánchez, Emilio Mariano Durán-Manuel, Claudia Camelia Calzada-Mendoza, Graciela Castro-Escarpulli, María Fernanda Quijano-Soriano, Liliana Nicolás-Sayago, Dulce Milagros Razo-Blanco Hernández, Marcela Villegas-Castañeda, Alejandro Cárdenas-Cantero, Mónica Alethia Cureño-Díaz, Marianela Paredes-Mendoza, Clemente Cruz-Cruz, Juan Manuel Bello-López

**Affiliations:** 1División de Investigación, Hospital Juárez de México, Mexico City 07760, Mexico; 2Laboratorio de Investigación Clínica y Ambiental, Departamento de Microbiología Escuela Nacional de Ciencias Biológicas, Instituto Politécnico Nacional, Mexico City 11340, Mexico; 3Facultad de Estudios Superiores Iztacala, UNAM, Tlalnepantla de Baz 04510, Mexico; 4Sección de Estudios de Posgrado, Escuela Superior de Medicina, Instituto Politécnico Nacional, Mexico City 11340, Mexico; 5Hospital Regional de Alta Especialidad “Bicentenario de la Independencia”, ISSSTE, Tultitlán de Mariano Escobedo 54916, Mexico; 6Dirección de Investigación, Hospital Juárez de México, Mexico City 07760, Mexico; 7División de Tecnología Ambiental, Universidad Tecnológica de Nezahualcóyotl, Nezahualcóyotl 57000, Mexico

**Keywords:** *Neisseria gonorrhoeae*, urogenital gonorrhea, epidemiology, Mexico, COVID-19

## Abstract

In Mexico, urogenital gonorrhea (UG) is one of the main sexually transmitted diseases notifiable by health systems around the world. Epidemiological data on sexually transmitted infections (STIs) in Mexico indicated that UG was “under control” until 2017. However, international epidemiological reports indicate the increase in incidence due to several factors, including an increase during the first year of the COVID-19 pandemic. These factors suggest that this phenomenon may occur in developing countries, including Mexico. Therefore, the aim of this study was to analyze national surveillance data on UG from 2003–2019 and the first year of the COVID-19 pandemic. An epidemiological study of cases and incidence of UG (2003–2020) was performed in the annual reports issued by the General Directorate Epidemiology in Mexico. Cases and incidence were classified and analyzed by year, sex, age group, and seasons (by temperature). Distribution of UG was carried out using heat maps for the whole country. Ultimately, a seasonal and correlation analysis was performed for UG cases versus temperature. The results showed that the distribution of cases and incidence by sex showed that there was no variation over 14 years. From 2016 onward, a significant increase in UG was observed before the pandemic. During the first year of the pandemic, a significant increase was observed in females aged 24–44 years. A heterogeneous distribution of UG was identified; however, border states were ranked among the top states with elevated incidences and cases. Lastly, the occurrence of UG was associated with temperature, related to summer. The information presented is intended to be useful to promote prevention and to contribute to visualize the distribution of UG over the last 18 years for decision making, and to show one of the consequences of the collapse of epidemiological surveillance of UG during the first year of the COVID-19 pandemic.

## 1. Introduction

Urogenital gonorrhea (UG) is a sexually transmitted infection (STI) caused by the bacterium *Neisseria gonorrhoeae*, whose reservoir is exclusively human; according to the World Health Organization (WHO) and other reports, it is one of the three main STIs [1,2]. This STI is widely distributed throughout the world and has been studied from various aspects considering the host–parasite relationship, as well as clinical aspects, diagnostics, and treatments; lately, it has been studied from the epidemiological point of view due to the emergence of multidrug-resistant strains and the behavior of its incidence during the current COVID-19 pandemic [3,4,5,6,7,8].

In Mexico, UG is one of the main bacterial sexually transmitted diseases and a public health problem, which has been associated with demographic characteristics, migratory phenomena, etc., as primary factors in the spread of the disease. The influence of immigration status in sex worker populations at the US border on the acquisition of STI (HIV, syphilis, gonorrhea, and chlamydia) has been analyzed. Studies have shown that migratory status influences the acquisition of STIs [9]. Conversely, it has been recognized that the decrease in the use of contraceptive methods, little fear of contracting an STI (including HIV), change in sexual behavior, lack of timely medical care, and the isolation of *N. gonorrhoeae* strains resistant to antibiotics are causes that favor the increase in cases of UG [10]. Lastly, cross-sectional surveys in developed countries have identified other risk factors in the acquisition of STIs, e.g., the use of drugs such as noninjectable methamphetamines [11]. 

In the local context, according to the Mexican Official Standard NOM-039-SSA2-2014 “*For prevention and control of sexually transmitted infections*”, UG is classified in the group of infections with a predominantly sexual mode of transmission, which excludes Reiter’s syndrome, HIV, and nongonococcal urethritis/cervicitis [12]. Furthermore, the identification of new cases of UG is nationally notifiable according to the Mexican Official Standard NOM-017-SSA2-2012 “*For epidemiological surveillance”* [13]. In Mexico, the diagnosis of UG is performed in the clinic and laboratory, through the identification of purulent secretions accompanied by urethritis and cervicitis with a history of 2–7 days after sexual intercourse. The observation of typical intracellular diplococci with Gram-negative stain affinity and complementary tests are mandatory in NOM-039-SSA2-2014. However, in the health systems of Mexican states with scarce resources, it is not known whether they are implemented. In developed countries, e.g., Spain, compulsory confirmatory diagnosis of UG can be made from tests via isolation of the causative agent through culture of secretions, DNA in situ hybridization, or molecular analysis by PCR. This is in accordance with the System of Obligatory Notifiable Diseases and the Microbiological Information System included in the National Epidemiological Surveillance Network (RENAVE in Spanish) [14]. 

Regarding STI epidemiology, new reports indicate that STIs, such as UG and syphilis, increased during the first year of the COVID-19 pandemic [15,16,17]. A 2020 US STI epidemiological surveillance report showed that gonorrhea cases increased significantly compared to 2019 [18]. These findings suggest that, since health efforts were focused on the COVID-19 pandemic, *N. gonorrhoeae* infections increased dramatically due to poor surveillance and control. Important factors have been recognized as contributing to the increase in UG during the COVID-19 pandemic. For example, in developed countries such as the USA, it was reported that, during the first year of the COVID-19 pandemic, most public and private laboratories capable of testing for STIs switched to SARS-CoV-2 detection tests [18,19]. 

Many suspected UG patients avoided attending clinics and hospitals during the pandemic, thus decreasing opportunities for diagnosis and treatment of this and many other infections [20]. This led to the spread of this and many other conditions; consequently, incidence rates and numbers of cases increased. Alternatively, empirical antimicrobial treatment of UG due to a lack of timely diagnosis opens possibilities for the emergence of multidrug-resistant strains, adding another problem of global importance, i.e., antimicrobial resistance [21]. Therefore, it is necessary to have epidemiological evidence that shows the behavior of UG by identifying risk groups, analyzing the geographical distribution, identifying new cases, and highlighting the possible association of environmental factors with the behavior of the disease throughout the Mexican Republic. 

The aim of this study was to analyze national surveillance data on UG from 2003–2019 and the first year of the COVID-19 pandemic through a descriptive, observational, and retrospective study. The information presented is intended to be useful to promote prevention and contribute to visualize the distribution of UG over the last 18 years (from 2003 to 2020) for decision making, and to show one of the consequences of the collapse of epidemiological surveillance of UG during the first year of the COVID-19 pandemic.

## 2. Materials and Methods

### 2.1. Period Analyzed of Urogenital Gonorrhea (UG)

The present study analyzed the incidence and number of cases of UG during 18 years (2003 to 2020) in the habitants of the Mexican Republic. Epidemiological data for the year 2021 from the Mexican Epidemiological Surveillance System (MESS) were not yet available on the website consulted. Regulatory policies in Mexico established mandatory reporting of UG cases in all 32 states of the country. This is according to the Official Mexican Standard “*For epidemiological surveillance*” (NOM-017-SSA2-2012) [13].

### 2.2. Operational Definition of a UG Case

In 2021, the General Directorate of Epidemiology (DGE in Spanish) issued the manual of “operational definitions of diseases subject to surveillance”, which states that the operational definition of a UG case is one where symptoms in women appear 2–5 days after infection, such as pain and burning during urination, painful sexual intercourse, severe lower abdominal pain and fever (if the infection spreads to the fallopian tubes), abnormal uterine bleeding, bleeding after sexual intercourse, and abnormal greenish, yellowish or foul-smelling discharge. In contrast, in men, symptoms may appear 1 month later and include pain and burning with urination, increased urinary frequency or urgency, white, yellow, or green discharge, red or swollen opening of the penis (urethra), and tender or swollen testicles [22]. With the characteristics described above, cases are divided into two groups:

*Probable cases*—patients presenting purulent discharge from the urethra or mild cervicitis accompanied by dysuria, 2–7 days after sexual intercourse.

*Confirmed cases*—any probable cases in which Gram staining shows typical Gram-negative intracellular diplococci in the secretions [12,22]. 

Interestingly, the criteria for identifying confirmed cases of UG conflict between NOM-039-SSA2-2014 (which includes microbiological culture or molecular tests) and the criteria of the manual of operational definitions of diseases subject to surveillance (which only includes diplococcal observation by Gram staining). For the purpose of this work, confirmed cases were based on NOM-039-SSA2-2014, considering that the year of publication of the manual of operational definitions of diseases subject to surveillance was 1 year after the first year of the COVID-19 pandemic (2021) [12].

### 2.3. Data Collection

The collection and presentation of the information were carried out under the observance of the principles of confidentiality and discretion indicated by the Federal Law on Accountability and Access to Public Government Information. Epidemiological data were taken from the morbidity yearbooks available at www.epidemiologia.salud.gob.mx/anuario/html/anuarios.html (accessed on 1 November 2022). The data reported on this website were previously generated and analyzed by the Ministry of Health through its SUAVE web platform at www.sinave.gob.mx (accessed on 1 November 2022). 

### 2.4. Epidemiological Analysis of UG

With the data on cumulative new cases and incidence by year and sex, an analysis of UG incidence (per 100,000 habitants) was carried out. This was used to construct an epidemic curve showing the behavior of “cases versus incidence” (both by year and by sex). To identify populations susceptible (by age) to *N. gonorrhoeae* infection, they were stratified into 11 age groups (0 to 65 years and over) and sex. Since epidemiological surveillance of the UG in the <1, 1–4 and 5–9 age groups was stopped in 2013, two analyses were conducted: a 10 year analysis in the 1–9 year age group and an 18 year analysis for the remainder of the population. With the information of national incidence, a distribution map was generated (per 100,000 habitants) during the period analyzed. For this purpose, four groups were generated by quartile classification that included the Mexican states from the lowest to the highest incidences (quartiles Q1, Q2, Q3, and Q4). According to this information, the Mexican territory was geographically mapped to identify the top states. 

### 2.5. Data Analysis

For the data analysis, the ANOVA test was employed to evaluate significant differences during the study period, according to incidence by state, cases and incidence by sex, and cases and incidence by age (*p* = 0.05). 

### 2.6. Seasonal Influence on UG Cases

A temporal analysis was performed with the variables “month/season of the year” versus “accumulated UG cases” in the study period. The temporal variation of the UG cases by temperature was analyzed using the Pearson test (<0.01, confidence = 95%, and error = 5%) using XLSTAT software. This was to confirm any related findings on the influence of annual seasonality and changes in ambient temperature in the Mexican territory. The variations in the national mean temperature were obtained from the monthly reports issued by *Comisión Nacional del Agua* (CONAGUA), available on the website https://smn.conagua.gob.mx/es/climatologia/temperaturas-y-lluvias/resumenes-mensuales-de-temperaturas-y-lluvias (accessed on 1 November 2022). Interpretation was performed using a Pearson rank-order correlation coefficient table. A *p*-value <0.05 was considered statistically significant.

## 3. Results

### 3.1. Behavior of New Cases and Incidence of UG by Sex

A total of 41,116 cases of UG were reported during the study period, with an annual average of 2284 cases. Comparison of the total number of cases and incidence (per 100,000 population) during the study period showed that the female sex had the highest number of cases compared to the male sex, with a ratio of 1.75, equivalent to 75% (1898 cases) for the female population. The distribution of cases and incidence by sex showed that neither epidemiological parameter showed significant changes from 2003 to 2013 (*p* = 0.05). Nevertheless, from 2016 onward, a significant increase was observed until 2020 (first pandemic year) with a final number of cases and incidences of 2505/4.87 and 4394/8.06 for the male and female sexes, respectively (*p* = 0.05). Figure 1 shows the epidemiological behavior of the number of cases and incidences of UG by sex during the period 2003–2020.

### 3.2. Susceptibility of UG per Age Groups and Sex

An analysis of UG cases was performed to determine which populations by age group and sex were most susceptible to infection. Cases of UG were reported in both sexes and in all age groups; however, this parameter increased dramatically in the age groups 15–19, 20–24, and 25–44 for both sexes. A maximum peak of cases was identified for the 25–44 age group in both males and females. In contrast, the results showed that the female sex was the most susceptible to acquiring this disease in the latter age group. A significant difference was observed between the 9927 and 8252 cases for males and females, respectively (*p* = 0.05). This difference was equivalent to 20.2% more cases for the female sex. For the remaining age groups, there was no significant difference in the number of cases by sex. From the 25–44 age group onward, a significant decrease in the number of total cases was observed. Lastly, because surveillance for UG was stopped from 2013 in age groups <1, 1–4, and 5–9, we could not identify changes in the number of cases after 2013; however, even with 10 year epidemiological surveillance, it was observed that age groups with lower UG were less susceptible to the development of UG. Figure 2 shows the distribution of cumulative UG cases by age group and sex over the 18 years analyzed. 

### 3.3. Geographical Distribution of Incidence of UG

To understand the geographic distribution of incidence of UG at national level, a heatmap of the epidemiological distribution of incidence was generated. 

This analysis allowed the integration of four quartiles (Q1, Q2, Q3, and Q4) made up of incidence ranges (from 0.25 to 7.26). In general, the top states by incidence (Q4: 4.86–7.26) were distributed mainly in the states located on the Mexican coasts and in the border area with Central America, except for Baja California Norte, located on the border with the United States. The states represented by the top quartile (Q4) were Baja California Norte, Zacatecas, Colima, Baja California Sur, Tabasco, Quintana Roo, Tamaulipas, and Nayarit. Furthermore, the quartile classification showed that Q1 was made up of states with incidences from 0.25 to 0.87, in contrast to Q2 from 1.10 to 2.35, and Q3 from 2.51 to 4.80. Figure 3 shows the national geographic distribution of incidence of UG per 100,000 habitants over 18 years in the Mexican Republic.

### 3.4. Distribution of UG Cases by Season and Temperature

The cumulative UG cases were distributed by month and were analyzed by distribution within the yearly seasons (winter, spring, summer, and autumn). In winter and autumn, the lowest numbers of cases were observed, with 2983 and 2551, respectively. A gradual increase of cases was identified, and the greatest number of UG cases was identified in summer (3945 cases). In Figure 4A, the distribution of UG cases per month/season during the study period is shown. Since an increase of UG cases was identified in the hottest months of the year (25.9 ± 0.66 °C), the possible relationship between distribution of cases and temperature was investigated. The information of new UG cases was studied by correlation analysis within the yearly temperature (minimum and maximum). The correlation analysis showed a positive correlation (R^2^ = 0.7129) that was statistically significant (<0.01, confidence = 95% and error = 5%) between the total number cases of UG and the average of the ambient temperature (Figure 4B). According to the Pearson test rank-order correlation coefficients, this correlation was “high” (0.6 ≤ r ≤ 0.79). 

## 4. Discussion

One of the objectives of national epidemiological surveillance of STIs is to generate strategies to implement control measures for their mitigation. These control measures, although national in scope, are primarily aimed at those populations that, due to diverse factors, may be more susceptible to the acquisition of STIs. Since UG is considered one of the most important STIs, it is necessary to show the general overview of this disease through its epidemiological behavior in a developing country such as Mexico. Moreover, the emergence of new diseases (such as COVID-19) may indirectly impact the epidemiology of other diseases, even if they are distantly related.

The global collapse of health systems due to the COVID-19 pandemic brought other consequences, including environmental quality, socioeconomic impact, management and governance, and transportation and urban design [23]. Recent reports indicate that the epidemiological surveillance of STIs has also been affected, as there has been a reduction in their diagnosis and treatment [16]. In the case of UG, it is one of the diseases that has been recognized in other countries as an epidemiological indicator of STI behavior during the COVID-19 pandemic, as it has shown an increase in the number of cases, reflecting the limited success of efforts to maintain social distancing [16].

Results from the epidemiology of UG in Mexico have shown that there was no significant variation in both sexes in the years prior to the COVID-19 pandemic; this behavior is similar to that reported by Kularatne et al. (2018) [24]. This work reported the analysis of 26 years of UG cases in the South African population (15–49 years of age), where it was shown that the rates did not show significant variation during the study period; however, it was identified that the female sex tends to be more susceptible to infection. This information shows that epidemiological surveillance and control measures have been successful in containing cases of UG. As shown in Figure 1, the number of cases and incidence increased significantly from 2016 in both sexes; nevertheless, it was observed that the female sex continues to be the group most susceptible to acquiring this infection. This finding on the susceptibility of women to infection has already been reported in the USA, where it is indicated that women are anatomically more susceptible to infection, wherebythe use of intrauterine devices and implants to prevent pregnancy and asymptomatic infections lead to increased incidence rates [25]. Conversely, it is not difficult to speculate the existence of infection events by undiagnosed male sexual partners, which could have an impact on the increase of cases and incidence in the female sex (Figure 1).

Failures in antimicrobial therapy have been recognized as a major cause of the spread of infectious diseases. UG is not the exception, as, due to the emergence of strains resistant to various antibiotics, such as cephalosporins and macrolides in the 2000s, the spread of antibiotic-resistant strains has been identified in various parts of the world [26,27]. We speculate that the increase in the number of cases detected since 2016 may be associated with the spread of antibiotic-resistant strains. This hypothesis is not so far from reality, as Russia reported a significant increase in the proportion of azithromycin-resistant *N. gonorrhoeae* isolates in the first 2 years of the pandemic (2020–2021). Genetic and epidemiological analysis of resistant isolates led to the conclusion that the emergence of azithromycin resistance in Russia in 2020 was the result of the spread of European strains of *N. gonorrhoeae* due to possible cross-border transfer [28].

National molecular epidemiological studies are needed to elucidate the presence and spread of resistant strains in cases of UG in the Mexican population, as only local studies have shown the presence of resistant strains of *N. gonorrhoeae*. Escobedo-Guerra et al. (2018) identified mutations in the *mtrR* gene, coding for a methyl tetrahydrofolate homocysteine methyltransferase reductase, which confers reduced susceptibility in *N. gonorrhoeae* strains [29]. 

This evidence published in the pre-pandemic era shows the possibility of the spread of resistant strains during the COVID-19 pandemic. Interestingly, in the first year of the pandemic, a dramatic increase in UG cases and incidence was observed in females by up to 75% compared to males. It has been reported that government strategies to manage the pandemic may influence the epidemiological behavior of STIs. Countries such as Sweden, Denmark, and Norway implemented a tool to track the stringency of government restrictions during the pandemic through “The Oxford COVID-19 Government Response Tracker” [30]. This study demonstrated success in reducing the number of reported cases in Sweden (17%) and Norway (39%) in 2020 compared to 2019, while, in Denmark, an increase of 21% was observed [31]. Moreover, active sexual life directly influences the detection of UG cases, however, as seen in the results shown in Figure 2, all age groups were susceptible to infection. 

The identification of cases in the population aged <1–14 years old suggests other possible ways of acquiring the infection. Transmission through sexual abuse has been recognized as the most frequent cause of infection among infants and children [32,33,34]. The 20–24 and 25–44 age groups were the most vulnerable to develop UG; women in the 25–44 age group were the most susceptible to infection. Contrasting this finding is the work reported by Yue et al. (2019), which showed that the rates (per 100,000 habitants) of infection among men and women aged 25–30 years were strikingly different, with rates of 5 and 25 for females and males, respectively [35]. A study on the epidemiological situation of gonorrhea in the Chilean population showed that the 20–24 age group had the highest infection rates, with the opposite being true for the Mexican population, where this same group was the second most susceptible (Figure 2) [36]. 

Analysis of the geographic distribution showed that UG incidence is not located in a particular region in the country; the states identified with high incidences were located in the south, north, and center of the whole country (Figure 3). Recent reports indicate the influence of border areas and the degree of urbanization on the incidence of STI cases [37,38]. In Mexico, border regions have already been recognized as a risk factor for HIV acquisition, where migration patterns, drug trafficking routes, and sex tourism contribute to the spread of STIs [39]. In the case of *Chlamydia* spp. and *N. gonorrhoeae* infections, risk factors associated with infection have been identified among sex workers in two Mexican border cities (Tijuana and Ciudad Juárez), located in states classified in Q4 and Q3 quartiles, respectively (Figure 3). 

In 2010, the “Healthy Border” initiative was established in the US border area, which includes 80 municipalities in six Mexican states and 48 counties in four US states [40]. The mission of the initiative established by the “United States–Mexico Border Health Commission” is to provide international leadership to optimize health and quality of life along the United States–Mexico Border, where STIs are included as diseases subject to surveillance by both countries. The detection of northern border states with high incidences of STIs in this work sets the stage for more robust surveillance efforts to contain STIs. In relation to the above, a prognostic analysis of HIV transmission behavior in border states of our country (some classified in the Q4 quartile of this work) has been carried out, and it is estimated that, within the next few years, 43.7% and 55.3% of new infections will be among men who have sex with men (MSM) and people who inject drugs, respectively, and that sex workers and their clients willconstitute <10% of new infections [41]. This reaffirms the fact that UG infections in these states are among the top regions in the entire country.

The identification of a maximum peak of UG cases in the summer months, where the average temperature in Mexico ranges between 25.9 ± 0.66 °C was the reason to perform a variable dependence correlation analysis (UG cases versus average temperature) (Figure 4A). The results showed a dependence of UG cases on temperature (Figure 4B). 

Comprehensive studies have been devoted to studying the seasonal behavior of STIs (associated with environmental temperature), where syphilis and gonorrhea are among the diseases that have been found to be temperature-dependent [42,43]. Furthermore, Cornelisse et al. (2017), through a seasonal cross-sectional analysis of sexual behavior and STIs in Melbourne, Australia, showed that cases of UG diagnoses in MSM were higher in summer than in winter [44]. There are possible reasons that could explain the seasonal behavior in the incidence of STIs. One is related to the seasonal fluctuation of sex hormones in warm months, which results in unsafe sexual encounters [42]. Lastly, the summer months are those with the highest tourist influx, which could also be related to this phenomenon. This pattern is similar to the findings of other studies, in which temporal variations by month in the behavior of STIs were observed [45,46]. 

Ultimately, weaknesses of the present study include the possible false-positive reporting of UG cases, which could be based on the clinical characteristics of patients (probable cases) with or without a differential diagnosis of urethritis or nongonococcal cervicitis [12,20]. Cases of recurrent reinfection of UG, as well as coinfections with other STI agents, may also be confounding factors in the epidemiology of this STI. Moreover, the identification of states with high average incidences of UG (Q4) may suffer a decrease in incidences within the studied period, which could lead to misclassifications. Future epidemiological studies will be conducted with the aim of analyzing the behavior of incidence at the state level using short temporalities, in order to verify our results.

## 5. Conclusions

The results presented in this study demonstrate relevant aspects about the control of UG in the Mexican territory in the pre-pandemic period, the possible reasons for the increase in cases and incidence in early 2016, and the possible impact of the COVID-19 pandemic on the collapse of health systems in Mexico in the timely detection of UG cases. From the above, we can also conclude that UG can be considered as an indicator of STI incidence during the COVID-19 pandemic in Mexico, and this marks the importance of monitoring cases in all health facilities in national crisis situations.

## Figures and Tables

**Figure 1 healthcare-11-02118-f001:**
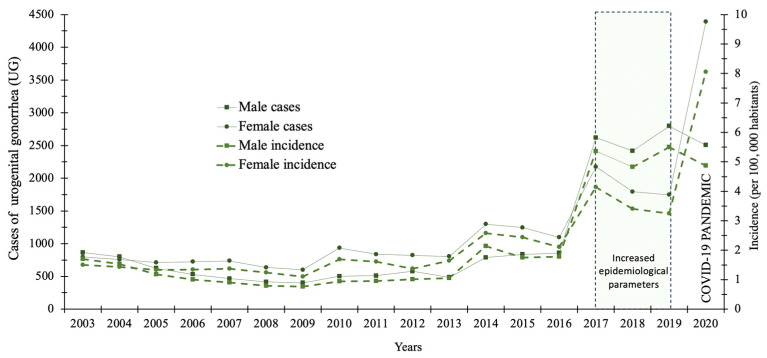
Epidemiological behavior of the total cases and incidence (per 100,000 habitants) of urogenital gonorrhea (UG) in Mexican territory from 2003 to 2020.

**Figure 2 healthcare-11-02118-f002:**
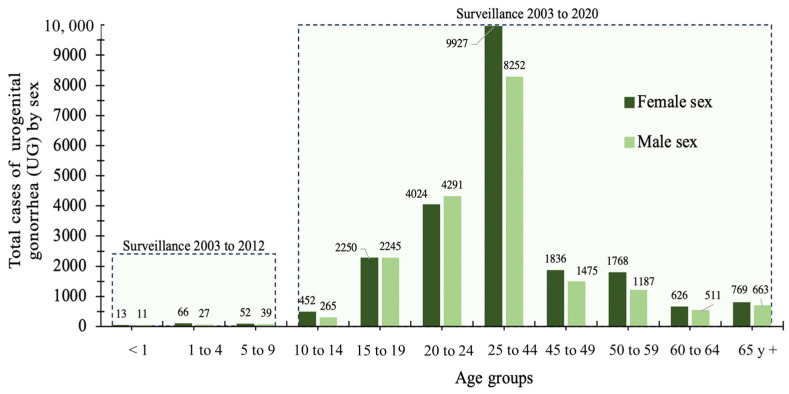
Accumulated cases of urogenital gonorrhea (UG) in Mexican territory by age groups and sex.

**Figure 3 healthcare-11-02118-f003:**
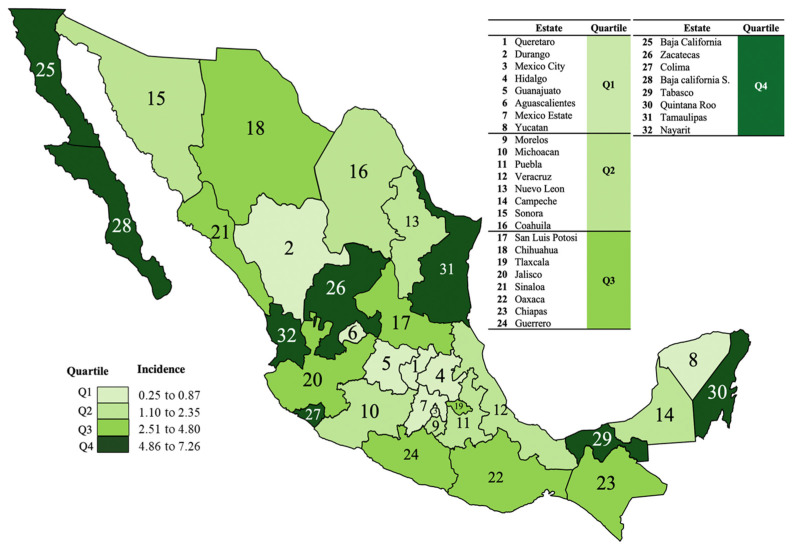
Geographic distribution by heat map of incidence rates (per 100,000 habitants) of urogenital gonorrhea (UG) by state in Mexican territory from 2003 to 2020.

**Figure 4 healthcare-11-02118-f004:**
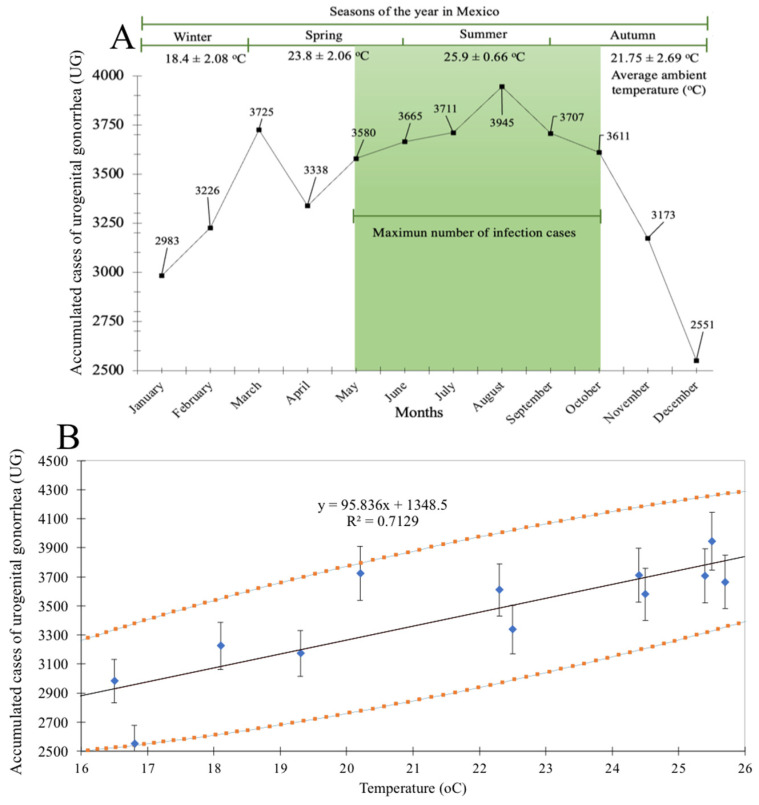
Seasonal influence by environmental temperature on the number of cases of urogenital gonorrhea (UG) in Mexico. (**A**) Seasonal variation of the total cases of UG in the period 2003–2020 in Mexican territory. (**B**) Correlation analysis between the total number cases of UG and temperature (°C).

## Data Availability

Bello-López, Juan Manuel (2023), “Urogenital gonorrhea in Mexico (2003–2020): impact of the COVID-19 pandemic on its epidemiological behavior”, Mendeley Data, V1, https://doi.org/10.17632/2r2jcxv88b.1.

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
