# Peer review of "Epidemiological Overview of Urogenital Gonorrhea in Mexico (2003–2020)"

_healthcare, 2023, doi:10.3390/healthcare11152118_

Round 1

Reviewer 1 Report

The article studies a very important topic, Neisseria gonorrhoeae infection, before and during the COVID-19 pandemic. Although there is a lot of information, the analysis and interpretation of the results can be substantially improved.

Introduction.

·         Line 50. It is suggested to use the term STI, which considers both symptomatic and asymptomatic infections.

·         Line 59. What demographic and economic factors have been associated with NG infection?, Have sexual behavior factors been evaluated?

·         Line 66-70. Laboratory confirmation is not required in Mexico. How is the diagnosis of GU made?

·         Line 78-79. “most public and private laboratories capable of testing for STDs switched  to COVID-19 testing”. This occurred in developed countries, but in Mexico before the pandemic, molecular detection of NG/CT was not performed. Detection is clinical, without laboratory confirmation.

·         Line 86-89. Is there information on environmental factors and their relationship with NG or other STIs?

Materials and methods

·         Line 107. “operational definitions of diseases subject to surveillance” need reference

·         Line 118. “Confirmed case”  The manual only mentions the Gram stain, on the other hand, the manual was published in 2021, after the period analyzed in the present study. In the data analyzed, no difference is made between a probable case and a confirmed case.

·         Line 116-118. Regarding the definition of probable case and confirmed case, the reference is missing.

·         Line 130. Since 2013, no cases have been reported in <1 year, 1-4 years and 5-9 years. It is necessary to consider this information.

·         Line 135. The maps contain cumulative incidence from 2003 to 2020 and the total number of cumulative cases from 2003 to 2020. What information do these maps provide? the aim of the study was to assess the effect of the COVID-19 pandemic. By combining all the information, the objective of the work is lost.

·         Line 139-140. Groups (A,B,C,D,E) and (1,2,3,4,5) were formed. How were these groups formed? quartiles? percentiles? Nothing is explained in methodology.

·         Line 142-143. Was the anova test used to assess the sex variable?

·         Line 145. Vector-borne diseases, diarrheal diseases or respiratory infections have a clear association with temperature. Why would an STI have some variation with temperature? No data is presented to suggest such an association with STIs, so there is a lack of support to carry out the analysis with temperature and seasons.

·         Line 150. Was the variation in temperatures for the entire country or for each of the states?

·         The study design is ecological, they have to be described based on that.

Results

·         Line 156. The number of new cases and the incidence, although they are different measures, obviously show the same trend through time or place. It is recommended to only use one of these parameters.

·         Line 163. “a significant increase was observed until 2020 with a final 163 number of cases and incidences of 2,505/4.87 and 4,394/8.06” How was this increase evaluated? What statistical test was used? what was the p-value?

·         Line 170. During the years 2003 to 2012 all age groups are reported, but since 2013 they stopped reporting: children under 1 year, 1-4 years and 5-9 years. The analysis must be carried out again.

·         Line 182. The accumulated number of cases by age group from 2003 to 2020 does not contribute to the objective of the study, which is to evaluate infection prior to and during the COVID-19 pandemic. In addition to the fact that in children, there is no information for all years.

·         Line 194. The aim of the study was to analyze the situation before and during the pandemic over time. Analyzing the maps with accumulated information does not show possible differences over time, so the objective of the study is not met.

·         Line 201. Only incident cases should be used for comparisons between states. New cases do not take into account the size of the population. That is the reason for the large differences between new cases and incidence on the maps.

·         Line 219. Similar to the previous observations, the objective is to study UG in relation to the COVID-19 pandemic. Once again, all the information is accumulated without considering the COVID-19 pandemic, which does not contribute to answering the research question.

·         Line 226. Using an average temperature for the entire country is inadequate, there is information at the state level. Mexico is a large country with large temperature variations

·         Line 231. Figure 4. In relation to the temperature and the months. Are the observations constant over the 18 years? Have there been changes during these 18 years? Why are there differences in relation to the season of the year? what is the hypothesis? What is the scientific support for this?

·         Line 231. Regarding the "R2" Was the correlation statistically significant? What is the slope of this correlation?

DISCUSSION

·         Line 239. “Recent reports indicate that epidemiological surveillance of STDs has also been affected, as there has been a reduction in their diagnosis and treatment. The article shows practically nothing about this, most of the analyzes were based on accumulated information, without considering the effect of COVID-19.

·         Line 280. Figure 2 does not show the difference between the pre-pandemic and pandemic periods, so it cannot be analyzed.

·         Line 282. Since 2013, cases of <1-14 years of age have ceased to be reported. It is important to mention this for the presentation of results and analysis.

·         Line 293-294. Geographic tropism? what does it mean ?

·         Line 296-297. The maps do not show statistically significant information that supports the hypothesis that the border states have a higher incidence, so this idea cannot be supported. Nor is it shown whether this information remains constant over the 18 years, since the accumulated number of cases does not have to be exactly the same as what is presented each year.

Author Response

Responses to reviewers

We appreciate the reviewer's thorough review of our manuscript, her/his thoughtful comments, and suggestions. We decided to revise our workand to answer point-by-point the reviewer’s queries and made the requested changes to our manuscript.

Reviewer 1

Introduction.

Comment 1: Line 50. It is suggested to use the term STI, which considers both symptomatic and asymptomatic infections.

Answer to comment 1: Line 50: The term “sexually transmitted disease (STD)” was modified as “sexually transmitted infection (STI)”. Additionally, in all manuscript the acronym “STD” was changed by “STI”.

Comment 2: Line 59. What demographic and economic factors have been associated with NG infection? Have sexual behavior factors been evaluated?

Answer to comment 2: Dear reviewer, we appreciate your valuable comment. Due to your observation, we decided to include additional information on risk factors for acquiring UG, as follows:

“The influence of immigration status in sex worker populations at the US border on the acquisition of STI (HIV, syphilis, gonorrhea, and chlamydia) has been analyzed. Studies have shown that migratory status influences the acquisition of STIs [9]. Conversely, it has been recognized that the decrease in the use of contraceptive methods, little fear of contracting an STI (including HIV), change in sexual behavior, lack of timely medical care, and the isolation of N. gonorrhoeae strains resistant to antibiotics, are causes that favor the increase in cases of UG [10]. Finally, cross-sectional surveys in developed countries have identified another risk factor in the acquisition of STIs, such as the use of drugs like non-injectable methamphetamines [11].

Additionally, the follow reference (11) was included, and the order of the references was modified according to the inclusion of the new reference.

11.Ye X, Li FR, Pan Q, Li Z, Yu GQ, Liu H, et al. Prevalence and associated factors of sexually transmitted infections among methamphetamine users in Eastern China: a cross-sectional study. BMC Infect Dis. 2022;22:7. http://doi: 10.1186/s12879-021-06987-8

Comment 3: Line 66-70. Laboratory confirmation is not required in Mexico. How is the diagnosis of GU made?

Answer to comment 3: Dear reviewer, we appreciate your valuable comment. Due to your observation, we decided to include additional information on diagnostic tests in Mexico, as follows:

“In Mexico, the diagnosis of UG is clinical and laboratory, through the identification of purulent secretions accompanied by urethritis and cervicitis with a history of two to seven days after sexual intercourse. The observation of typical intracellular diplococci with Gram-negative stain affinity, and complementary tests are mandatory in NOM-039-SSA2-2014. However, in the health systems of Mexican states with scarce resources, it is not known whether they are implemented. In developed countries, e.g., Spain, compulsory confirmatory diagnosis of UG can be made from tests based on isolation of the causative agent through culture of secretions, DNA in situ hybridization or molecular analysis by PCR. This is in accordance with the System of Obligatory Notifiable Diseases and the Microbiological Information System included in the National Epidemiological Surveillance Network (RENAVE, for its acronym in Spanish) [14].

Additionally, the follow reference (14) was included, and the order of the references was modified according to the inclusion of the new reference.

  1. Red Nacional de Vigilancia Epidemiológica. RENAVE. Protocolo de Vigilancia de Infección gonocócica (Available online https://www.isciii.es/QueHacemos/Servicios/VigilanciaSaludPublicaRENAVE/EnfermedadesTransmisibles/Documents/archivos%20A-Z/INFECCION%20GONOCOCICA/Protocolo%20de%20Vigilancia%20de%20Infección%20gonocócica.pdf (accesed on 31-05-2023)

Comment 4: Line 78-79. “Most public and private laboratories capable of testing for STDs switched to COVID-19 testing”. This occurred in developed countries, but in Mexico before the pandemic, molecular detection of NG/CT was not performed. Detection is clinical, without laboratory confirmation.

Answer to comment 4: Dear reviewer, we appreciate your valuable comment. Due to your observation, we decided modified the text, as follows:

“For example, early in the pandemic, most public and private laboratories capable of testing for STIs switched to COVID-19 testing [16,17].”

Was modified as:

“For example, in developed countries such as the USA, it was reported that during the first year of the COVID-19 pandemic most public and private laboratories capable of testing for STIs switched to SARS-CoV-2 detection tests [18,19]”.

Comment 5: Line 86-89. Is there information on environmental factors and their relationship with NG or other STIs?

Answer to comment 5: Dear reviewer, we appreciate your valuable comment. In relation to your comment, although the information is limited, there are some studies (including an article previously published by our work group, “reference 43”), which identify a relationship between temperature and the behavior of STIs. Additionally, we include in the final part of the manuscript additional information that strengthens our findings as follows.

There are possible reasons that could explain the seasonal behavior in the incidence of STIs. One is related to the seasonal fluctuation of sex hormones in warm months, which results in unsafe sexual encounters [42]. Finally, the summer months are the ones with the highest tourist influx, which could also be related to this phenomenon. This pattern is similar to the findings of other studies, in which temporal variations by month in the behavior of STIs were observed [45,46].

Additionally, we added two references [45 and 45]

  1. Fortenberry JD, Orr DP, Zimet GD, Blythe MJ. Weekly and seasonal variation in sexual behaviors among adolescent women with sexually transmitted diseases. J Adolesc Health. 1997;20:420-5. https://doi: 10.1016/S1054-139X(96)00275-3.

  1. Wright RA, Judson FN. Relative and seasonal incidences of the sexually transmitted diseases. A two-year statistical review. Br J Vener Dis. 1978;54:433-40. https://doi: 10.1136/sti.54.6.433.

Materials and methods.

Comment 6: Line 107. “Operational definitions of diseases subject to surveillance” need reference.

Answer to comment 6: Dear reviewer, we appreciate your valuable comment. “Operational definitions of diseases subject to surveillance” was reference (Reference 22) and the order of the references was modified according to the inclusion of the new reference.

  1. Vigilancia epidemiológica convencional de casos nuevos de enfermedad. Definiciones operacionales de enfermedades sujetas a vigilancia convencional. Secretaria de Salud/Subsecretaría de prevención y promoción de la salud/Dirección Adjunta de Epidemiología. Available online: https://epidemiologia.salud.gob.mx/gobmx/salud/documentos/manuales/DefinicionesOperacionales_Padecimientos_Sujetos_a_VE.pdf (accessed on 29-05-2023).

Comment 7: Line 118. “Confirmed case” The manual only mentions the Gram stain, on the other hand, the manual was published in 2021, after the period analyzed in the present study. In the data analyzed, no difference is made between a probable case and a confirmed case.

Answer to comment 7: Dear reviewer, we appreciate your valuable comment. Due to your observation, we decided deleted the follow text.

“Positive and intentional microbiological culture for N. gonorrhoeae, and alternative serological and molecular diagnostic tests such as antigen testing and PCR, respectively, are also considered valid diagnostic tools”.

Additionally, we decided to include the follow text:

Interestingly, the criteria for identifying confirmed cases of UG conflict between NOM-039-SSA2-2014 (which includes microbiological culture or molecular tests) and the criteria of the manual of operational definitions of diseases subject to surveillance (which only includes diplococcal observation by Gram staining). For the purpose of this work, confirmed cases were based on NOM-039-SSA2-2014, considering that the year of publication of the manual of operational definitions of diseases subject to surveillance was one year after the first year of the COVID-19 pandemic (2021) [12].

Comment 8: Line 116-118. Dear reviewer, we appreciate your valuable comment. Regarding the definition of probable case and confirmed case, the reference is missing.

Answer to comment 8: The reference “[20]” was included as follow:

  1. Vigilancia epidemiológica convencional de casos nuevos de enfermedad. Definiciones operacionales de enfermedades sujetas a vigilancia convencional. Secretaria de Salud/Subsecretaría de prevención y promoción de la salud/Dirección Adjunta de Epidemiología. Available online: https://epidemiologia.salud.gob.mx/gobmx/salud/documentos/manuales/DefinicionesOperacionales_Padecimientos_Sujetos_a_VE.pdf (accessed on 29-05-2023).

Comment 9: Line 130. Since 2013, no cases have been reported in <1 year, 1-4 years, and 5-9 years. It is necessary to consider this information.

Answer to comment 9: Dear reviewer, we appreciate your valuable comment. Due to your observation, we decided modified the Figure 2. Two epidemiological surveillance classifications by age groups were generated.

Due to the previous change, the following is modified:

Figure 2. Accumulated cases of gonococcal infection of the gonococcal urethritis (GU) in Mexican territory by age groups and sex from 2003 to 2020.

Was modified as follow:

Figure 2. Accumulated cases of gonococcal gonorrhea (GU) in Mexican territory by age groups and sex.

Also, In section 2.4. Epidemiological analysis of UG, the follow text was incuded:

“Since epidemiological surveillance of the UG in the <1, 1-4 and 5-9 year age groups were stopped in 2013, two analyses were conducted, a 10-year analysis in the 1-9 year age group and an 18-year analysis for the rest of the population”.

Also, In section 3.2. Susceptibility of UG per age groups and sex, the follow text was incuded:

Finally, because surveillance for UG was stopped from 2013 in age groups <1, 1-4 and 5-9 years old, we could not identify changes in the number of cases after 2013, howev-er, even with ten-year epidemiological surveillance it was observed that age groups with lower UG are less susceptible to the development of UG”.

Comment 10. Line 135. The maps contain cumulative incidence from 2003 to 2020 and the total number of cumulative cases from 2003 to 2020. What information do these maps provide? the aim of the study was to assess the effect of the COVID-19 pandemic. By combining all the information, the objective of the work is lost.

Answer to comment 10: Dear reviewer, we appreciate your valuable comment. Derived from his precise observation, we consider that we have overestimated the impact of COVID-19 in our manuscript (since it is only reflected in Figure 1). Therefore, we have made the decision to give the manuscript a different perspective of analysis. These changes allow showing a general epidemiological panorama of UG in Mexico and showing the possible impact that the pandemic had on the behavior of the epidemiological parameters (cases and incidence) of this disease in the Mexican population in 2020.

If you allow us this change in the perspective of the manuscript, Figure 1 may remain unchanged, and the other graphics presented with minor modifications. On the other hand, other changes are added to the complete document as shown below:

  • The title of manuscript:

“Urogenital Gonorrhea in Mexico (2003-2020): Impact of the COVID-19 Pandemic on its Epidemiological Behavior”

Was changed by:

“Epidemiological Overview of Urogenital Gonorrhea in Mexico (2003-2020)”

  • In the abstract and introduction sections, the follow sentences were included as follow:

Abstract: In Mexico, urogenital gonorrhea (UG) is one of the main sexually transmitted diseases notifiable by health systems around the world. Epidemiological data on sexually transmitted infections (STIs) in Mexico indicated that UG was “under control” until 2017. However, international epidemiological reports indicate the increase in incidence due to several factors, including a the increase…”

“Discussion: One of the objectives of national epidemiological surveillance of STIs is to generate strategies to implement control measures for their mitigation. These control measures, although national in scope, are primarily aimed at those populations that, due to di-verse factors, may be more susceptible to the acquisition of STIs. Since UG is considered one of the most important STIs, it is necessary to show the general overview of this dis-ease through its epidemiological behavior in a developing country such as Mexico. Moreover, the emergence of new diseases (such as COVID-19) may indirectly impact the epidemiology of other diseases, even if they are distantly related”.

Comment 11: Line 139-140. Groups (A, B, C, D, E) and (1, 2, 3, 4, 5) were formed. How were these groups formed? quartiles? percentiles? Nothing is explained in methodology.

Answer to comment 11: Since we generate incidence groups and cases arbitrarily, we decided to formally generate new groups based on quartiles (Q1, Q2, Q3 and Q4). In addition, we have considered your comment 15 (results section), and we have removed the case distribution map. Therefore, the following was modified:

2.4. Epidemiological analysis of UG, section was modified as follow:

With the information on total cases and national incidence, two distribution maps were generated, one assigned to the number of cumulative cases and the other to state incidence (per 100,000 inhabitants) during the period analyzed. For this purpose, five groups were generated that included the Mexican states from the highest to the lowest number of accumulated cases (groups 1, 2, 3, 4, and 5) and incidence (groups A, B, C, D, and E)”.

Was modified as follow:

“With the information of national incidence, a distribution map was generated (per 100,000 habitants) during the period analyzed. For this purpose, four groups were generated by quartile classification that included the Mexican states from the lowest to the highest incidences (quartiles Q1, Q2, Q3, and Q4)”.

A new section was included as:

2.5. Data analysis

For the data analysis, the ANOVA test was employed to evaluate significant dif-ferences during the study period, according to  incidence by state, cases and incidence by sex, and cases and incidence by age (p=0.05).

  • The follow text was deleted of Results secion:

“The first distribution map allowed the classification of Mexican states according to the number of cases (from 75 to 3,813) into 5 groups (1, 2, 3, 4 and 5). In general, the largest number of cases were distributed mainly in the states located on the Mexican coast and in the border area with Central America, except for Baja California Norte, located on the border with the United States. The results showed that 17.42% (7,183 cases) were attributed to the states of Jalisco (Central Pacific) and Tamaulipas (Northern Gulf), considered the coastal states with the highest number of cases (cluster 5).

The remaining number of cases (34,032) were distributed in 4 groups (1, 2, 3, and 4) in the following order: group 1 (1.1%/465 cases), group 2 (13.34%/5,501 cases), group 3 (30.05%/ 15,684 cases), and group 4 (29.99%/12,361 cases). The geographical distribution of cumulative UG cases over eighteen years is shown in Figure 3A. Furthermore, to understand the possible correlation between the national distribution of UG cases and the distribution of state incidences, a second distribution heat map was generated by using the national average incidences (per 100,000 inhabitants) as a variable”.

  • The full section “3. Geographical distribution of cases and incidence of UG”:

“To understand the geographic distribution of cases and incidence of UG at the national level, heat maps of the epidemiological distribution of cases and incidence were generated. This analysis allowed the integration of five clusters (A, B, C, D, and E) made up of incidence ranges (from 0.25 to 7.26). This evaluation showed a redistribution of top states initially classified by number of cases except for the state of Tamaulipas, which remained in the top E group with the highest incidences (6.47-7.26). This group incorporated other states, such as Tabasco and Quintana Roo (states bordering Central America) and Nayarit (Central Pacific state). Group D concentrated some Gulf of Mexico states (Chiapas, Guerrero, Baja California, Colima, and Baja California Sur) with incidences between 3.10-5.31. The state of Zacatecas (located in the center of the country) was also part of group D. Group C states (Sonora, Coahuila, San Luis Potosi, Chihuahua, Tlaxcala, Jalisco, Sinaloa, and Oaxaca) had incidences between 2.20-2.95. The last groups (A and B) had incidences between 0.25-0.87 and 1.10-1.99, respectively. Figure 3 shows the national geographic distribution of cumulative UG cases (Fig. 3A) and national state UG incidences (Fig. 2B) over 18 years in the Mexican Republic”.

Was changed as follow:

3.3. Geographical distribution of incidence of UG

To understand the geographic distribution of incidence of UG at national level, a heat map of the epidemiological distribution of incidence was generated.

This analysis allowed the integration of four quartiles (Q1, Q2, Q3, and Q4) made up of incidence ranges (from 0.25 to 7.26). In general, the top states by incidence (Q4: 4.86-7.26), were distributed mainly in the states located on the Mexican coasts and in the border area with Central America, except for Baja California Norte, located on the border with the United States. The states represented by the top quartile (Q4) were Baja California Norte, Zacatecas, Colima, Baja California Sur, Tabasco, Quintana Roo, Tamaulipas, and Nayarit. Furthermore, the quartile classification showed that Q1 was made up of states with incidences from 0.25 to 0.87, Q2 from 1.10 to 2.35, and Q3 from 2.51-4.80. Figure 3 shows the national geographic distribution of incidence of UG per 100, 000 habitants over 18 years in the Mexican Republic.

  • In discussion section, the follow text:

“Analysis of the geographic distribution of UG showed that there is no geographic tropism associated with this disease; nevertheless, states with high numbers of cases and incidences were identified in states located in the south and north of the country (Figure 3). Recent reports indicate the influence of border areas and the degree of urbanisation on the incidence of STI cases [34,35]. In Mexico, border regions have already been recognised as a risk factor for HIV acquisition, where migration patterns, drug trafficking routes and sex tourism contribute to the spread of STIs [36]. In the case of Chlamydia spp. and N. gonorrhoeae infections, risk factors associated with infection have been identified among sex workers in two Mexican border cities (Tijuana and Ciudad Juárez), located in groups C and D, respectively (Figure 3)”.

Was modified as follow:

Analysis of the geographic distribution showed that the UG incidence is not located in a particular region in the country; the states identified with high incidences were located in the south, north, and center of the whole country (Figure 3). Recent reports indicate the influence of border areas and the degree of urbanization on the incidence of STI cases [38,39]. In Mexico, border regions have already been recognized as a risk factor for HIV acquisition, where migration patterns, drug trafficking routes, and sex tourism contribute to the spread of STIs [40]. In the case of Chlamydia spp. and N. gonorrhoeae infections, risk factors associated with infection have been identified among sex workers in two Mexican border cities (Tijuana and Ciudad Juárez), located in states classified in Q4 and Q3 quartiles, respectively (Figure 3).

Comment 12: Line 142-143. Was the anova test used to assess the sex variable?

Answer to comment 12: Dear reviewer, we appreciate your valuable comment.

The follow sentence: “For data analysis, the ANOVA test was employed to evaluate significant differences during the study period, number of cases by state, sex, and age (p=0.05)”

Was modified as follow:

For the data analysis, the ANOVA test was employed to evaluate significant differences during the study period, according to incidence by state, cases and incidence by sex, and cases and incidence by age (p=0.05). A new section was included as: “2.5 Data analysis”.

Comment 13: Line 145. Vector-borne diseases, diarrheal diseases or respiratory infections have a clear association with temperature. Why would an STI have some variation with temperature? No data is presented to suggest such an association with STIs, so there is a lack of support to carry out the analysis with temperature and seasons.

Answer to comment 13: Dear reviewer, we appreciate your valuable comment. We have included a sentence that justifies the analysis of the correlation between the environmental temperature (reflected in the seasons of the year) and the number of UG cases. The follow sentence was included (2.6 Seasonal influence on UG cases):

“The above to confirm any related findings on the influence of annual seasonality and changes in ambient temperature in the Mexican territory”.

In relation to your comment, although the information is limited, there are some studies (including an article previously published by our work group, reference 43), which identify a relationship between temperature and the behavior of STIs. Additionally, previously published information was included in the discussion section, which demonstrates the seasonal influence on the biological and behavioral behavior of the population against STIs.

The follow information was included:

“There are possible reasons that could explain the seasonal behavior in the incidence of STIs. One is related to the seasonal fluctuation of sex hormones in warm months, which results in unsafe sexual encounters [42]. Finally, the summer months are the ones with the highest tourist influx, which could also be related to this phenomenon. This pattern is similar to the findings of other studies, in which temporal variations by month in the behavior of STIs were observed [45,46]”.

Additionally, we added two references [41 and 42]

  1. Fortenberry JD, Orr DP, Zimet GD, Blythe MJ. Weekly and seasonal variation in sexual behaviors among adolescent women with sexually transmitted diseases. J Adolesc Health. 1997;20:420-5. https://doi: 10.1016/S1054-139X(96)00275-3.

  1. Wright RA, Judson FN. Relative and seasonal incidences of the sexually transmitted diseases. A two-year statistical review. Br J Vener Dis. 1978;54:433-40. https://doi: 10.1136/sti.54.6.433.

Comment 14: Line 150. Was the variation in temperatures for the entire country or for each of the states?

Answer to comment 14: Dear reviewer, we appreciate your valuable comment. “In 2.5. Seasonal influence on UG cases”, it is indicated that the national average temperature is as follows:

“The variations in the national mean temperature were obtained from the monthly reports….”

Results.

Comment 15: Line 156. The number of new cases and the incidence, although they are different measures, obviously show the same trend through time or place. It is recommended to only use one of these parameters.

Answer to comment 15: Dear reviewer, we appreciate your valuable comment. We kindly ask you to review the response of comment 10, hoping to satisfy your request.

Comment 16: Line 163. “a significant increase was observed until 2020 with a final 163 number of cases and incidences of 2,505/4.87 and 4,394/8.06” How was this increase evaluated? What statistical test was used? what was the p-value?

Answer to comment 16: Dear reviewer, we appreciate your valuable comment. P value was included (p=0.05). In 2.5 Data analysis section, statistical test is indicated (ANOVA).

Comment 17: Line 170. During the years 2003 to 2012 all age groups are reported, but since 2013 they stopped reporting: children under 1 year, 1-4 years and 5-9 years. The analysis must be carried out again.

Answer to comment 17: Dear reviewer, we appreciate your valuable comment. We kindly ask you to review the response of comment 9, hoping to satisfy your request.

Comment 18: Line 182. The accumulated number of cases by age group from 2003 to 2020 does not contribute to the objective of the study, which is to evaluate infection prior to and during the COVID-19 pandemic. In addition to the fact that in children, there is no information for all years.

Answer to comment 18: Dear reviewer, we appreciate your valuable comment. We kindly ask you to review the response of comments 9 and 10, hoping to satisfy your request.

Comment 19: Line 194. The aim of the study was to analyze the situation before and during the pandemic over time. Analyzing the maps with accumulated information does not show possible differences over time, so the objective of the study is not met.

Answer to comment 19: Dear reviewer, we appreciate your valuable comment. We kindly ask you to review the response of comments 9, 10 and 11, hoping to satisfy your request.

Comment 20: Line 201. Only incident cases should be used for comparisons between states. New cases do not consider the size of the population. That is the reason for the large differences between new cases and incidence on the maps.

Answer to comment 20: Dear reviewer, we appreciate your valuable comment. We kindly ask you to review the response of comments 11, hoping to satisfy your request.

Comment 21: Line 219. Similar to the previous observations, the objective is to study UG in relation to the COVID-19 pandemic. Once again, all the information is accumulated without considering the COVID-19 pandemic, which does not contribute to answering the research question.

Answer to comment 21: Dear reviewer, we appreciate your valuable comment. We kindly ask you to review the response of comments 9, 10 and 11, hoping to satisfy your request.

Comment 22: Line 226. Using an average temperature for the entire country is inadequate, there is information at the state level. Mexico is a large country with large temperature variations.

Answer to comment 22: Dear reviewer, we appreciate your valuable comment. However, the decision to consider the national average temperature as a variable in the correlation analysis with the cases, is based on previous works that are referred to in the “discussion” section (including an article previously published by our research group(reference 43)). On the other hand, since the correlation between total cases versus temperature was analyzed, we cannot make use of state temperatures, since it was not related to state incidence values, but rather to the number of cases. Finally, as you can see in figure 4, the standard deviations of the national average temperatures are reliable because they do not exceed the national average.

Comment 23: Line 231. Figure 4. In relation to the temperature and the months. Are the observations constant over the 18 years? Have there been changes during these 18 years? Why are there differences in relation to the season of the year? what is the hypothesis? What is the scientific support for this?

Answer to comment 23: Dear reviewer, we appreciate your valuable comment. We kindly ask you to review the response of comment 5 hoping to satisfy your request.

Comment 24: Line 231. Regarding the "R2" Was the correlation statistically significant? What is the slope of this correlation?

Answer to comment 24: Dear reviewer, we appreciate your valuable comment. Derived from your observation, we decided to improve the results related to the correlation analysis as follows:

The sentence:

  • (In materials and methods section): The temporal variation of the GU cases by temperature was analysed by using the general linear model of repeated measures with the Bonferroni adjustment (P<0.05) by using the SPSS v23 software.

Was modified as follow:

  • (In 2.6 Materials and methos section): The temporal variation of the UG cases by temperature was analyzed by using the Pearson test (<0.01, confidence=95% and error=5%) using XLSTAT software.

The sentence:

  • (In results section): The correlation analysis shows a positive correlation (R2= 0.7074) between the total number cases of UG and the average of the ambient temperature (Figure 4B). According to Spearman Rank-Order Correlation Coefficients, this correlation is very strong”.

Was modified as follow:

  • The correlation analysis shows a positive correlation (R2= 0.7129) statistically significant (<0.01, confidence=95% and error=5%) between the total number cases of UG and the average of the ambient temperature (Figure 4B). According to Pearson test Rank-Order Correlation Coefficients, this correlation is “high correlation” (0.6 ≤ r ≤ 0.79).

Finally, the figure 4 was modified and slope value was included (into full equation of trend behavior)

Discussion

Comment 25: Line 239. “Recent reports indicate that epidemiological surveillance of STDs has also been affected, as there has been a reduction in their diagnosis and treatment. The article shows practically nothing about this, most of the analyzes were based on accumulated information, without considering the effect of COVID-19.

Answer to comment 25:  Dear reviewer, we appreciate your valuable comment. We kindly ask you to review the response of comments 9, 10 and 11, hoping to satisfy your request.

Comment 26: Line 280. Figure 2 does not show the difference between the pre-pandemic and pandemic periods, so it cannot be analyzed.

Answer to comment 26: Dear reviewer, we appreciate your valuable comment. We kindly ask you to review the response of comments 9, 10 and 11, hoping to satisfy your request.

Comment 27: Line 282. Since 2013, cases of <1-14 years of age have ceased to be reported. It is important to mention this for the presentation of results and analysis.

Answer to comment 27: Dear reviewer, we appreciate your valuable comment. We kindly ask you to review the response of comment 9, hoping to satisfy your request.

Comment 28: Line 293-294. Geographic tropism? what does it mean?

Answer to comment 28: The follow text: “Analysis of the geographic distribution of UG showed that there is no geographic tropism associated with this disease; nevertheless, states with high incidences were identified in states located in the south and north of the country (Figure 3).”

Was modified as follow:

“Analysis of the geographic distribution showed that the UG incidence is not located in a particular region in the country; the states identified with high incidences were located in the south, north, and center of the whole country (Figure 3)”.

Comment 29: Line 296-297. The maps do not show statistically significant information that supports the hypothesis that the border states have a higher incidence, so this idea cannot be supported. Nor is it shown whether this information remains constant over the 18 years, since the accumulated number of cases does not have to be exactly the same as what is presented each year.

Answer to comment 29: Dear reviewer, we appreciate your valuable comment. We kindly ask you to review the response of comment 11, hoping to satisfy your request. Additionally, we decide include a “weakness part” in discussion section as follow:

“Ultimately, weaknesses of the present study include the possible false positive reporting of UG cases, which could be based on the clinical characteristics of patients (probable cases) with or without a differential diagnosis of urethritis or non-gonococcal cervicitis [12, 20]. Cases of recurrent reinfection of UG, as well as co-infections with other STI agents may also be a confounding factor in the epidemiology of this STI. Moreover, the identification of states with high average incidences of UG (Q4) may suffer a decrease in incidences within the studied period, which could lead to misclassifications. Future epidemiological studies will be conducted with the aim of analyzing the behavior of incidence at the state level using short temporalities, in order to verify our results”.

Reviewer 2 Report

1. Result section (3.2) Susceptibility of GU per age groups and sex: As shown in figure 2, more cases are seen in female sex (9,927) than male sex (8252). However, the author described 20.2 % more cases for the male sex, while describing. Which is correct?  

2. The author speculated that the increase in the number of cases detected since 2016 due to the spread of antibiotic resistant strains. However, figure 1 shows drastic increase in the number of cases in both sexes during 2020. Is there any evidence of spreading antibiotic resistant strains during 2020? 

3. Since the study was intended/ focused on urogenital gonorrhea and its epidemiological behavior after the introduction of COVID-19 (2020); however, there is not enough information/reasons regarding drastic increase in the number of cases during 2020. It is better to generate another/independent figure that shows incidence of age, months, and geographic distribution.      

Author Response

Reviewer 2

Comment 1. Result section (3.2) Susceptibility of GU per age groups and sex: As shown in figure 2, more cases are seen in female sex (9,927) than male sex (8252). However, the author described 20.2 % more cases for the male sex, while describing. Which is correct? 

Answer to comment 1. Dear reviewer, we appreciate your valuable comment. The follow text:

“This difference is equivalent to 20.2% more cases for the male sex”.

Was modified as follow:

“This difference is equivalent to 20.2% more cases for the female sex”.

Comment 2: The author speculated that the increase in the number of cases detected since 2016 due to the spread of antibiotic resistant strains. However, figure 1 shows drastic increase in the number of cases in both sexes during 2020. Is there any evidence of spreading antibiotic resistant strains during 2020? 

Answer to comment 2: Dear reviewer, we appreciate your valuable comment. Derived from his observation, we decided to include the following paragraph as follow:

“This hypothesis is not so far from reality, as Russia reported a significant increase in the proportion of azithromycin-resistant N. gonorrhoeae isolates in the first two years of the pandemic (2020-2021). Genetic and epidemiological analysis of resistant isolates led to the conclusion that the emergence of azithromycin resistance in Russia in 2020 was the result of the spread of European strains of N. gonorrhoeae due to possible cross-border transfer [29]”.

Additionally, the follow reference was included (29), and the order of references was modified:

  1. Kandinov, I., Dementieva, E., Filippova, M., Vinokurova, A., Gorshkova, S., Kubanov, A., ... & Gryadunov, D. (2023). Emergence of Azithromycin-Resistant Neisseria gonorrhoeae Isolates Belonging to the NG-MAST Genogroup 12302 in Russia.Microorganisms,11(5), 1226. https://doi.org/10.3390/microorganisms11051226

Comment 3: Since the study was intended/ focused on urogenital gonorrhea and its epidemiological behavior after the introduction of COVID-19 (2020); however, there is not enough information/reasons regarding drastic increase in the number of cases during 2020. It is better to generate another/independent figure that shows incidence of age, months, and geographic distribution.

Answer to comment 3: Dear reviewer, we appreciate your valuable comment. Dear reviewer, we appreciate your valuable comment. Derived from his precise observation, we consider that we have overestimated the impact of COVID-19 in our manuscript (since it is only reflected in Figure 1). Therefore, we have made the decision to give the manuscript a different perspective of analysis. These changes allow showing a general epidemiological panorama of UG in Mexico and showing the possible impact that the pandemic had on the behavior of the epidemiological parameters (cases and incidence) of this disease in the Mexican population in 2020.

If you allow us this change in the perspective of the manuscript, Figure 1 may remain unchanged, and the other graphics presented with minor modifications. On the other hand, other changes are added to the complete document as shown below:

  • The title of manuscript:

“Urogenital Gonorrhea in Mexico (2003-2020): Impact of the COVID-19 Pandemic on its Epidemiological Behavior”

Was changed by:

“Epidemiological Overview of Urogenital Gonorrhea in Mexico (2003-2020)”

  • In the abstract section, the follow sentence was included as follow:

Abstract: In Mexico, urogenital gonorrhea (UG) is one of the main sexually transmitted diseases notifiable by health systems around the world. Epidemiological data on sexually transmitted infections (STIs) in Mexico indicated that UG was “under control” until 2017. However, international epidemiological reports indicate the increase in incidence due to several factors, including a the increase…”

“Discussion: One of the objectives of national epidemiological surveillance of STIs is to generate strategies to implement control measures for their mitigation. These control measures, although national in scope, are primarily aimed at those populations that, due to di-verse factors, may be more susceptible to the acquisition of STIs. Since UG is considered one of the most important STIs, it is necessary to show the general overview of this dis-ease through its epidemiological behavior in a developing country such as Mexico. Moreover, the emergence of new diseases (such as COVID-19) may indirectly impact the epidemiology of other diseases, even if they are distantly related”.

Reviewer 3 Report

 Many thanks for performing this analysis and sharing your data with scientific community.

I have read the aforementioned manuscript with interest. This is an epidemiological study of gonorrhea covering a whole country of Mexico. Besides, it reports epidemiology between the years 1993-2009. This is a huge amount of data and information which makes this study a suitable candidate for publication. 

Line 158 national average of …. should be annual average of…

This study has no association with the covid pandemic but the authors focused on the collapse of sexual health services after the pandemic. Here the authors tried to attract the attention of scientific community hoping that their manuscript will worth much in the context of COVID19. 

In the discussion section authors have focused on possible resistance of NG without providing a robust clue in the results section. 

To improve the manuscript authors may consider removing all COVID and resistance related arguments and focus on their findings. Moreover, they may consider to discuss their findings by citing a paper about HIV epidemiology in Mexico. So that it would be nice to see if the both STD epidemiology overlap. Moreover, it would be nice if the authors provide some clues of recurrent or reinfections in their cohort.

Moreover, in addition to providing gender and age epidemiology in a single table, they may consider to dissect their data in 5-year pentiles (ie 1999-2004, 2005-2010 vice versa). Lower risk of perceived conception leads to condomless sex among elder population which lead to higher detection of primary HIV infection in elder population.  It would be nice to see if this could be the case for gonorrhea.

please provide limitations and strength of your data.

Author Response

Reviewer 3

Comment 1: Line 158 national average of …. should be annual average of…

Answer to comment 1: Dear reviewer, we appreciate your valuable comment. The follow text:

“A total of 41,116 cases of GU were reported during the study period, with a national average of 2,284 cases.”

Was modified as follow:

“A total of 41,116 cases of UG were reported during the study period, with an annual average of 2,284 cases”.

Comment 2: This study has no association with the covid pandemic, but the authors focused on the collapse of sexual health services after the pandemic. Here the authors tried to attract the attention of scientific community hoping that their manuscript will worth much in the context of COVID19. 

Answer to comment 2: Dear reviewer, we appreciate your valuable comment. Dear reviewer, we appreciate your valuable comment. Dear reviewer, we appreciate your valuable comment. Derived from his precise observation, we consider that we have overestimated the impact of COVID-19 in our manuscript (since it is only reflected in Figure 1). Therefore, we have made the decision to give the manuscript a different perspective of analysis. These changes allow showing a general epidemiological panorama of UG in Mexico and showing the possible impact that the pandemic had on the behavior of the epidemiological parameters (cases and incidence) of this disease in the Mexican population in 2020.

If you allow us this change in the perspective of the manuscript, Figure 1 may remain unchanged, and the other graphics presented with minor modifications. On the other hand, other changes are added to the complete document as shown below:

  • The title of manuscript:

“Urogenital Gonorrhea in Mexico (2003-2020): Impact of the COVID-19 Pandemic on its Epidemiological Behavior”

Was changed by:

“Overview of Urogenital Gonorrhea in Mexico (2003-2020)”

  • In the abstract section, the follow sentence was included as follow:

Abstract: In Mexico, urogenital gonorrhea (UG) is one of the main sexually transmitted diseases notifiable by health systems around the world. Epidemiological data on sexually transmitted infections (STIs) in Mexico indicated that UG was “under control” until 2017. However, international epidemiological reports indicate the increase in incidence due to several factors, including a the increase…”

“Discussion: One of the objectives of national epidemiological surveillance of STIs is to generate strategies to implement control measures for their mitigation. These control measures, although national in scope, are primarily aimed at those populations that, due to di-verse factors, may be more susceptible to the acquisition of STIs. Since UG is considered one of the most important STIs, it is necessary to show the general overview of this dis-ease through its epidemiological behavior in a developing country such as Mexico. Moreover, the emergence of new diseases (such as COVID-19) may indirectly impact the epidemiology of other diseases, even if they are distantly related”.

Comment 3: In the discussion section authors have focused on possible resistance of NG without providing a robust clue in the results section.

Answer to comment 3: Dear reviewer, we appreciate your valuable comment. Indeed, we do not have evidence on the emergence of resistant strains, however we corrected this deficiency through hypotheses and information from other countries on the impact of antimicrobial resistance in Neisseria gonorrheae through the following text:

“This hypothesis is not so far from reality, as Russia reported a significant increase in the proportion of azithromycin-resistant N. gonorrhoeae isolates in the first two years of the pandemic (2020-2021). Genetic and epidemiological analysis of resistant isolates led to the conclusion that the emergence of azithromycin resistance in Russia in 2020 was the result of the spread of European strains of N. gonorrhoeae due to possible cross-border transfer [29]”.

Additionally, the follow reference was included (29), and the order of references was modified:

  1. Kandinov, I., Dementieva, E., Filippova, M., Vinokurova, A., Gorshkova, S., Kubanov, A., ... & Gryadunov, D. (2023). Emergence of Azithromycin-Resistant Neisseria gonorrhoeae Isolates Belonging to the NG-MAST Genogroup 12302 in Russia.Microorganisms,11(5), 1226. https://doi.org/10.3390/microorganisms11051226

Comment 4: To improve the manuscript authors may consider removing all COVID and resistance related arguments and focus on their findings. Moreover, they may consider discussing their findings by citing a paper about HIV epidemiology in Mexico. So that it would be nice to see if both STD epidemiology overlap. Moreover, it would be nice if the authors provide some clues of recurrent or reinfections in their cohort.

Answer to comment 4: Dear reviewer, we appreciate your valuable comment. We kindly ask you to review the response of comment 1 hoping to satisfy your request. And the other hand, we decided include information about HIV incidence in state borders in Mexico to show the impact of this disease although measures of control has been implemented in state borders in our country.

The follow was included in Discussion section:

“In relation to the above, a prognostic analysis of HIV transmission behavior in border states of our country (some classified in the Q4 quartile of this work) has been carried out, and it is estimated that within the next few years, 43.7% and 55.3% of new infections will be among men who have sex with men (MSM) and people who inject drugs, respectively, and sex workers and their clients constitute <10% of new infections [42]. This reaffirms the fact that UG infections in these states are among the top regions in the entire country.

Additionally, we included the follow reference (42):

  1. Fraser H, Borquez A, Stone J, Abramovitz D, Brouwer KC, Goodman-Meza D, Hickman M, et al. Overlapping Key Populations and HIV Transmission in Tijuana, Mexico: A Modelling Analysis of Epidemic Drivers. AIDS Behav. 2021;25:3814-3827. doi: 10.1007/s10461-021-03361-2.

Finally, we decided include a weakness part of this study in discussion section to talk about recurrent reinfections of UG as possible confounding factor in data showed.

The text was included as follow:

“Ultimately, weaknesses of the present study include the possible false positive re-porting of UG cases, which could be based on the clinical characteristics of patients (probable cases) with or without a differential diagnosis of urethritis or non-gonococcal cervicitis [12, 20]. Cases of recurrent reinfection of UG, as well as co-infections with other STI agents may also be a confounding factor in the epidemiology of this STI. More-over, the identification of states with high average incidences of UG (Q4) may suffer a decrease in incidences within the studied period, which could lead to misclassifications. Future epidemiological studies will be conducted with the aim of analyzing the behavior of incidence at the state level using short temporalities, in order to verify our results”.

Comment 5: Moreover, in addition to providing gender and age epidemiology in a single table, they may consider dissecting their data in 5-year pentiles (i.e., 1999-2004, 2005-2010 vice versa). Lower risk of perceived conception leads to condomless sex among elder population which lead to higher detection of primary HIV infection in elder population.  It would be nice to see if this could be the case for gonorrhea.

Answer to comment 5: Dear reviewer, we appreciate your valuable comment. Derived from the valuable observation of another reviewer, we identified that the epidemiological surveillance of the UG was stopped for certain age groups, in such a way that we modified figure 2 (classification by age groups). His recommendation would impact the structure of much of the manuscript. In addition, we do not have the necessary information for the suggested exercise. We highly value his observation, however, please review the enormous substantial changes that our document underwent and that undoubtedly improved the quality of the manuscript.

Comment 6: Please provide limitations and strength of your data.

Answer to comment 6: Dear reviewer, we appreciate your valuable comment. We kindly ask you to review the response of comment 4 hoping to satisfy your request.

Reviewer 4 Report

Thank you for sharing such a well written and insightful article with the healthcare community. I would like to make few suggestions as a clinician which could help the readers understand the health scenarios in Mexico which the article portrays:

1. Gonococcal urethritis in most cases occur as a co-infection with chlamydia infection, and that is the main reason clinicians often treat both infection in patients with clinical symptoms of gonorrhea. Therefore, I will suggest the authors includes that in this article.

2. since GU is a form of STD/STI, I suggest the authors include a data on the trend of other STD (syphilis, HIV) and STI (Herpes, chlamydia, trichomoniasis, etc) during pre-COVID vs COVID-19 era, as most patients often present with co-infections.

3. Lastly, any explanation to why females tend to have higher incidence than the male? could it be a case of re-infection from untreated male partners?

Author Response

Revisor 4:

Comment 1. Gonococcal urethritis in most cases occur as a co-infection with chlamydia infection, and that is the main reason clinicians often treat both infection in patients with clinical symptoms of gonorrhea. Therefore, I will suggest the authors includes that in this article.

Answer to comment 1: Dear reviewer, we appreciate your valuable comment. Without a doubt, your suggestion is very valuable to us. However, as you can see, the manuscript underwent significant changes in structure, content, analysis, and approach. Including variables such as other ITS would generate a new restructuring of the manuscript, including the presentation of a new proposal. Therefore, please review the changes presented in this new revised version hoping it meets your expectations.

Comment 2: since GU is a form of STD/STI, I suggest the authors include a data on the trend of other STD (syphilis, HIV) and STI (Herpes, chlamydia, trichomoniasis, etc.) during pre-COVID vs COVID-19 era, as most patients often present with co-infections.

Answer to comment 2: Dear reviewer, we appreciate your valuable comment. The National Epidemiological Surveillance System of Mexico does not have data on cases of coinfection or reinfection by STIs. However, we have enriched the discussion section, where we address the behavior of other STIs in our country, as well as a section on weaknesses regarding reinfection events. Therefore, please review the changes presented in this new revised version hoping it meets your expectations.

Comment 3: Lastly, any explanation to why females tend to have higher incidence than the male? could it be a case of re-infection from untreated male partners?

Answer to comment 3: Dear reviewer, we appreciate your valuable comment. Derived from your comment, we enrich the discussion section where the finding of the susceptibility of the female sex to acquire UG is addressed.

The follow text was modified as follow:

“This finding on the susceptibility of women to infection has already been reported in the USA, where it is indicated that women are anatomically more susceptible to infection, the use of intrauterine devices and implants to prevent pregnancy and asymptomatic infections lead to increased incidence rates [26]. Conversely, it is not difficult to speculate the existence of infection events by undiagnosed male sexual partners, which could have had an impact on the increase of cases and incidence in the female sex (Figure 1)”.

Finally, we decided to include a section on weaknesses of the study. This section mentions the possible confusion in the diagnosis of GU in the population and the impact of the year of publication of the manual of “operational definitions of diseases subject to surveillance”.

The follow text was included:

Ultimately, weaknesses of the present study include the possible false positive reporting of UG cases, which could be based on the clinical characteristics of patients (probable cases) with or without a differential diagnosis of urethritis or non-gonococcal cervicitis [12, 20]. Cases of recurrent reinfection of UG, as well as co-infections with other STI agents may also be a confounding factor in the epidemiology of this STI. Moreover, the identification of states with high average incidences of UG (Q4) may suffer a decrease in incidences within the studied period, which could lead to misclassifications. Future epidemiological studies will be conducted with the aim of analyzing the behavior of incidence at the state level using short temporalities, in order to verify our results.

Round 2

Reviewer 2 Report

The revised version of the manuscript is now acceptable for publication.